# FINISH FIRST, PERFECT LATER: TEST-TIME TOKEN-LEVEL CROSS-VALIDATION FOR DIFFUSION LARGE LANGUAGE MODELS

## ABSTRACT

Diffusion large language models (dLLMs) have recently emerged as a promising alternative to autoregressive LLMs, offering accelerated parallel decoding and improved global context modeling through bidirectional attention. However, vanilla decoding strategies in dLLMs suffer from a critical limitation: once a token is accepted, it can no longer be revised in subsequent steps. As a result, early mistakes persist across iterations, harming both intermediate predictions and final output quality. To address this issue, we propose TOLERATOR (**To**ken-**Le**vel **Cr**oss-V**a**lida**t**ion **R**efinement), a training-free decoding strategy that leverages cross-validation among predicted tokens. Unlike existing methods that follow a single progressive unmasking procedure, TOLERATOR introduces a two-stage process: (i) sequence fill-up and (ii) iterative refinement by remasking and decoding a subset of tokens, while treating the remaining ones as context. This design enables previously accepted tokens to be reconsidered and corrected when necessary, leading to more reliable diffusion decoding outputs. We evaluate TOLERATOR on five standard benchmarks covering language understanding, code generation, and mathematics. Empirically, our method achieves consistent improvements over the baselines under the same computational budget. These findings suggest that decoding algorithms are crucial to realizing the full potential of diffusion large language models [1].

## 1 INTRODUCTION

Large language models (LLMs) (Chowdhery et al., 2022; Hurst et al., 2024; Comanici et al., 2025) have driven remarkable progress across diverse NLP domains (Zhao et al., 2023; Minaee et al., 2024). The dominant architecture behind these advances is the autoregressive (AR) transformer (Vaswani et al., 2017). While highly effective, AR decoding is inherently sequential, creating a fundamental bottleneck that limits generation parallelism (Fu et al., 2024; Xia et al., 2024).

To address this, diffusion language models (Austin et al., 2021a; Li et al., 2022) have emerged as a powerful alternative, generating sequences through iterative denoising with bidirectional attention and parallel token predictions. This paradigm offers distinct advantages over AR models (Li et al., 2025b), including accelerated inference, stronger global coherence, and controllable quality–speed trade-offs. Recent progress (Labs et al., 2025; Nie et al., 2025; Ye et al., 2025b) has further demonstrated the practicality and competitiveness of scaled diffusion large language models (dLLMs). Commercial dLLMs such as Mercury Coder (Labs et al., 2025) and Gemini Diffusion (Google DeepMind, 2025) claim to match autoregressive LLMs (Hurst et al., 2024; Team et al., 2024) in performance while achieving up to $10\times$ faster inference speed on tasks, like code generation (Chen et al., 2021; Austin et al., 2021b).

Despite recent advances, current dLLM decoding strategies (Israel et al., 2025; Yu et al., 2025; Wu et al., 2025) suffer from a critical limitation: once a token is predicted and accepted, it is typically fixed and cannot be modified in later steps (Wang et al., 2025; von Rütte et al., 2025). For instance, in two widely adopted open-source dLLMs, LLaDA (Nie et al., 2025) and Dream (Ye et al., 2025b),

---

[1]Code and data are anonymously available.

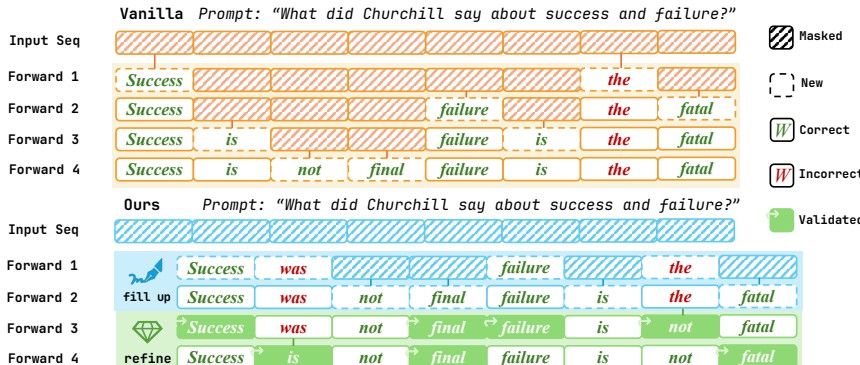

Figure 1: **Overview of TOLERATOR**. Compared to the vanilla decoding strategy, we first fill the masked tokens with high parallelism and then iteratively refine the draft through token-level cross-validation. Here, cross-validation means tokens alternately act as the target and the context of prediction. This process allows previously accepted tokens to be revisited and corrected when necessary.

a token is considered *accepted* if, at a specific iteration, it is unmasked and no longer remasked, as illustrated in Figure 1. Once accepted, it will serve as fixed context for all future predictions. This causes early mistakes to persist and propagate throughout the generation process (Wang et al., 2025; von Rütte et al., 2025).

There have been some early explorations on this issue. ReMDM (Wang et al., 2025) introduces a sampler that applies a stochastic backward remasking process for predicted tokens. RCR (He et al., 2025) tracks each token's running max confidence and remasks persistently low-confidence tokens. GIDD (von Rütte et al., 2025) trains diffusion models with a mixing schedule that interpolates between data and noise distributions to enable the remasking of predicted tokens. While these works demonstrate the significance of dLLM decoding strategy, their improvements have not achieved ideal performance on general tasks, so the challenge remains an open problem.

To further bridge this gap, we propose TOLERATOR (**To**ken-**Le**vel **Cr**oss-**V**alida**t**ion **R**efinement), a test-time dLLM decoding method that explicitly separates generation into two stages: *fill-up* and *refinement*. In the first stage, we fill up the masked tokens following vanilla dLLM decoding strategy. In the second stage, we iteratively refine this draft by remasking and decoding subsets of tokens while using the remaining ones as context, so that predictions are revised by cross-validating against one another. This two-stage process allows previously accepted tokens to be revisited multiple times and corrected when necessary. Our approach differs from existing strategies which perform refinement within the ongoing generation process. By explicitly decoupling fill-up and refinement into two separate phases, TOLERATOR enables a more thorough form of token-level error correction than prior methods.

We evaluate TOLERATOR on five standard benchmarks across language understanding (TriviaQA (Joshi et al., 2017), GPQA (Rein et al., 2024)), code generation (MBPP (Austin et al., 2021b), HumanEval (Chen et al., 2021)), and mathematics (GSM8K (Cobbe et al., 2021)). We use vanilla decoding, ReMDM (Wang et al., 2025), and RCR (He et al., 2025) as baselines. Experimental results show that, under the same computational cost measured by the number of forward steps, TOLERATOR achieves noticeable and consistent improvements over the baselines (relatively improve 17.9% for Dream (Ye et al., 2025b) and 15.3% on LLaDA (Nie et al., 2025)). We further conduct ablation studies and analyze the characteristics of our method. Qualitative studies further highlight how cross-validation corrects errors in practice. Overall, these findings confirm that decoding strategy is not merely an implementation detail, but a crucial factor that substantially influences the performance of dLLMs.

## 2 RELATED WORK

### 2.1 FROM AUTOREGRESSION TO DIFFUSION

Modern natural language generation (Hendrycks et al., 2020; Suzgun et al., 2023; Rein et al., 2024) has been dominated by the autoregressive (AR) model architecture like GPT (Brown et al., 2020) and

LLaMA (Touvron et al., 2023). Despite its empirical success, AR models introduce a fundamental bottleneck: generation is inherently sequential, limiting decoding parallelism (Li et al., 2023; Zou et al., 2023). To address this limitation, diffusion language models (Austin et al., 2021a; Li et al., 2022) have emerged as a promising alternative (Li et al., 2025b). By reversing a noising process over multiple steps, diffusion language models generate tokens in parallel (Labs et al., 2025) while leveraging full bidirectional attention (Nie et al., 2025; Ye et al., 2025b).

Existing diffusion language models can be classified into three main categories depending on how the diffusion process is applied. Early *continuous diffusion language models* (Li et al., 2022; Strudel et al., 2022; Karimi Mahabadi et al., 2024; Lovelace et al., 2023; Dieleman et al., 2022) denoised *embeddings* before mapping them back to tokens. However, this paradigm struggles with issues like optimization and has largely been replaced by discrete diffusion language models. *Discrete diffusion language models* (Austin et al., 2021a; He et al., 2023) define diffusion directly in *token* space, and further scale up model parameter size (Gong et al., 2025), achieving the state-of-the-art with open-source models like Dream (Ye et al., 2025b) and LLaDA (Nie et al., 2025). A third line integrates AR philosophy with diffusion, including block-wise or multi-level scheduling (Han et al., 2023; Wu et al., 2023) and the reintroduction of sequential dependency while retaining diffusion-style refinement (Arriola et al., 2025; Huang & Tang, 2025).

## 2.2 TRAINING AND INFERENCE STRATEGIES IN DIFFUSION LANGUAGE MODELS

Beyond architectural explorations, another line of work studies how to effectively train diffusion LMs. Large-scale instruction tuning (Ye et al., 2025b; Nie et al., 2025), has demonstrated that diffusion models can achieve general capabilities comparable to autoregressive LLMs. Researchers explore refinements of the training objective: simplified masked losses (Shi et al., 2024; Sahoo et al., 2024), likelihood-based formulations (Gulrajani & Hashimoto, 2023), and variants that enhance generation robustness and reasoning (von Rütte et al., 2025; Ye et al., 2025a). Another direction focuses on adapting reinforcement learning to diffusion, either to strengthen reasoning (Huang & Tang, 2025; Ye et al., 2024; Zhao et al., 2025) or for preference optimization (Zhu et al., 2025).

Decoding is another key bottleneck for diffusion language models: parallel generation improves efficiency but often degrades quality. Adaptive Parallel Decoding (APD) (Israel et al., 2025) mitigates this trade-off by adjusting the degree of parallelism with an auxiliary autoregressive verifier, while dilated (Luxembourg et al., 2025) scheduling further accelerate inference. At the same time, KV-caching (Ma et al., 2025; Wu et al., 2025) and autoregressive-guided unmasking (Hu et al., 2025) are applied to further accelerate dLLMs. Recent work also addresses flexibility (Li et al., 2025a; Kim et al., 2025) by extending diffusion to variable-length and token insertion.

## 2.3 ERROR CORRECTION IN DIFFUSION DECODING

It is often claimed that vanilla diffusion language models possess an inherent ability for error correction, since each position is repeatedly predicted as the context evolves over iterations (Li et al., 2023; 2025b). However, this view is incomplete: once a token is accepted, it becomes fixed and cannot be revised. For example, LLaDA (Nie et al., 2025) and Dream (Ye et al., 2025b) decide at every iteration whether a token should be further remasked; if it is not, the token is considered accepted and remains unchanged thereafter. As a result, any early mistake will persist and propagate through subsequent steps, limiting the reliability of diffusion generation.

Several methods have sought to address this limitation. ReMDM (Wang et al., 2025) introduces a probabilistic remasking process that allows already revealed tokens to be re-predicted. RCR (He et al., 2025) proposes a simple confidence-based strategy that remasks uncertain tokens during inference. GIDD (von Rütte et al., 2025) modifies the corruption process with hybrid noise at the training time. While these approaches demonstrate the feasibility of token revision, their empirical gains remain relatively modest on general tasks or they require additional training, leaving the core problem unresolved. In contrast, our approach departs from prior work by explicitly decoupling fill-up and refinement. We first generate a draft following vanilla diffusion decoding, and then apply a targeted refinement stage that revisits the accepted tokens according to a cross-validation principle. This separation not only makes error correction conceptually more systematic but also delivers markedly stronger empirical gains.

# 3 METHODOLOGY

## 3.1 PRELIMINARIES

**Decoding in dLLMs.** We consider the decoding process of discrete diffusion large language models (Ye et al., 2025b; Nie et al., 2025). Specifically, let $x_i^{(t)} \in \mathcal{V}$ denote the token at position $i \in \{1, \dots, L\}$ and time step $t \in \{0, \dots, T\}$, where $\mathcal{V}$ is the vocabulary, $L$ is the sequence length, and $T$ is the total number of forward steps. At inference time, the sequence is initialized with

$$x^{(0)} = \big(c_1, \dots, c_m, \underbrace{[\text{MASK}]_{m+1}, \dots, [\text{MASK}]_L}_{L-m}\big) \in \mathcal{V}^L,$$

where $c_1$ to $c_m$ are prompt tokens and the remaining $L - m$ positions are masked tokens. At each time step, the diffusion large language models output the logits of all masked tokens and decode them by sampling, where $y_i^{(t)} \sim p_\theta(\cdot \mid x^{(t)}, t)$, and $p_\theta$ is the conditional distribution parameterized by the dLLMs. A deterministic rule then decides whether to accept or remask each decoded token. Specifically, the next sequence is constructed as

$$x_i^{(t+1)} = \begin{cases} y_i^{(t)}, & \text{accepted}, \\ [\text{MASK}], & \text{remasked}, \end{cases} \quad \text{for } i \notin \mathcal{I}_t, \qquad x_j^{(t+1)} = x_j^{(t)} \quad \text{for } j \in \mathcal{I}_t.$$

where $\mathcal{I}_t \subseteq \{1, \dots, L\}$ is the index set of tokens already accepted at step $t$. In the vanilla setup, each step accepts approximately $\lfloor L/T \rfloor$ tokens, which are selected based on criteria like model confidence or entropy. Different dLLMs may adopt alternative decoding strategies; for example, semi-autoregressive decoding (Nie et al., 2025) only proceeds to the next block once all tokens in the current block have been accepted. Our study focuses on the vanilla setup, as it is widely adopted in existing dLLMs.

**Limitations of Conventional dLLM Decoding.** In this conventional setup, masked positions are iteratively refined, while accepted tokens become fixed and remain unchanged. Formally, once a position index $i$ enters the visible set $\mathcal{I}_t$, we have $i \in \mathcal{I}_{t'}$ and $x_i^{(t')} = x_i^{(t)}$ for all $t' > t$. As a result, an early error at position $j \in \mathcal{I}_t$ is permanently preserved and enters the context for all future predictions $p_\theta(x_i^{(t')} \mid x^{(t'-1)}, t' - 1)$, where $i \notin \mathcal{I}_{t'-1}$. Such errors cannot be revised and may propagate through the decoding process as persistent noise, ultimately degrading the quality of the generated sequence.

## 3.2 METHOD OVERVIEW

To overcome this limitation, we propose TOLERATOR (**To**ken-**Le**vel C**r**oss-**V**alida**t**ion **R**efinement), which moves beyond the traditional view of decoding as a single, progressively unmasking trajectory, and instead reframes it as a two-stage process of *fill-up* and *refinement*.

**Stage I (Sequence Fill-Up).** In the fill-up stage, the model produces a coarse draft by filling masked positions following vanilla dLLM decoding strategy, providing a complete but potentially imperfect hypothesis of the output.

**Stage II (Cross-Validation Refinement).** In the refinement stage, our iterative procedure follows a token-level cross-validation principle, where tokens alternately act as validator and as validation targets. This alternating role improves overall consistency of generated sequence.

This design offers a training-free, model-agnostic and effective solution to the challenge of irreversible early errors and their propagation in the decoding process.

## 3.3 SEQUENCE FILL-UP

The sequence fill-up stage is largely based on the vanilla dLLM decoding procedure described in Section 3.1. To facilitate the refinement stage, we introduce a modification: the logit penalty on the End-of-Text (EoT) token.

**EoT penalty.**   Since the refinement stage can correct errors, we prefer longer and more informative drafts rather than overly short completions. To this end, we apply an *EoT penalty* (Bai et al., 2021; Laban et al., 2020), which discourages generation of EoT tokens in the fill-up stage. Concretely, we scale down the logit of the EoT token by a factor $\lambda_{\text{eot}} > 1$ before softmax. While this adjustment does not directly improve draft quality, it effectively prevents early termination and produces drafts that are better suited for subsequent refinement. Formally, let $z_v$ be the unnormalized logit for token $v$ at position $i$ and time step $t$. The penalized distribution is

$$\tilde{p}_\theta(v \mid x^{(t)}, t) \propto \begin{cases} \exp(z_v)/\lambda_{\text{eot}}, & \text{if } v = [\text{EoT}] \\ \exp(z_v), & \text{otherwise.} \end{cases}$$

Finally, the fill-up stage produces a sequence consisting of the prompt tokens and model predictions for previously masked positions:

$$x^{(\rho T)} = \left( c_1, \ldots, c_m, x_{m+1}^{(\rho T)}, \ldots, x_L^{(\rho T)} \right) \in \mathcal{V}^L,$$

where $x_i^{(\rho T)} \neq [\text{MASK}]$ for all $i > m$. Here $\rho \in (0, 1)$ controls the split between the two stages.

### 3.4 Cross-Validation Refinement

The refinement stage corrects errors in the draft with a token-level cross-validation principle, where tokens alternately act as validator and as validation targets. In each iteration, a subset of tokens is *sampled*, *remasked* and *decoded* conditioned on the preserved context, progressively reducing mistakes and improving coherence.

**Iterative Refinement.**   At each iteration $k$, we remask a random subset $S^{(k)} \subseteq \{m+1, \ldots, L\}$ of non-prompt positions, sampled at rate $\gamma_k$ so that $|S^{(k)}| = \lfloor \gamma_k(L-m) \rfloor$.

$$x_i^{(k)} = \begin{cases} [\text{MASK}], & i \in S^{(k)} \\ x_i^{(k)}, & \text{otherwise.} \end{cases}$$

The sequence for the next iteration is then obtained by predicting the masked tokens:

$$x_i^{(k+1)} = \begin{cases} y_i^{(k)}, & i \in S^{(k)} \\ x_i^{(k)}, & \text{otherwise,} \end{cases} \quad \text{where } y_i^{(k)} \sim p_\theta(\cdot \mid x^{(k)}, k).$$

In each iteration, a subset of generated tokens is held fixed as context, while the remaining tokens are remasked and decoded to better align with them. Iterating this process gradually improves the coherence of the entire sequence.

**Annealed Refinement Rate.**   To improve the stability of refinement steps, we anneal the refinement rate $\gamma_k$ over time. Higher refinement rates in early iterations encourage broader corrections of initial errors, while lower rates in later iterations help stabilize the predictions. We adopt a cosine annealing schedule with both upper and lower bounds:

$$\gamma_k = \gamma_{\min} + \tfrac{1}{2}(\gamma_{\max} - \gamma_{\min})\left(1 + \cos\left(\tfrac{\pi k}{K}\right)\right),$$

where $k$ is the current refinement iteration and $K$ is the total number of refinement steps.

## 4 Experiment

### 4.1 Experimental Setup

**Models.**   Following previous studies (Ma et al., 2025; Israel et al., 2025; Wu et al., 2025; He et al., 2025), we evaluate our method on two representative open-source dLLMs: **Dream-v0-Instruct-7B** (Ye et al., 2025b) and **LLaDA-8B-Instruct** (Nie et al., 2025). Both of them are state-of-the-art representatives of open-source discrete diffusion large language models.

**Datasets & Metrics.** To assess the general effectiveness of our method, we evaluate it on three representative tasks with five standard benchmarks: (i) language understanding with **TriviaQA** (Joshi et al., 2017) and **GPQA** (Rein et al., 2024), (ii) code generation with **HumanEval** (Chen et al., 2021) and **MBPP** (Austin et al., 2021b), and (iii) mathematics with **GSM8K** (Cobbe et al., 2021). We report accuracy for TriviaQA, GPQA, GSM8K and pass@1 for HumanEval and MBPP.

**Baselines.** We compare our method against the vanilla decoding strategy and two training-free baselines that propose to revise the accepted tokens. (i) **Vanilla** strategy follows the vanilla dLLM decoding procedure, where once a token is accepted, it remains fixed throughout the generation process and cannot be revised. (ii) **ReMDM** (Wang et al., 2025) introduces a stochastic sampler that applies a backward remasking process for predicted tokens. (iii) **RCR** (He et al., 2025) records each token's running max confidence and remasks persistently low-confidence tokens.

**Configurations.** For fairness, all methods are evaluated with the same dLLM backbones with the same total number of forward passes in the zero-shot setting. We also equalize the computational cost between baselines and our method by allocating the same total forward-step budget to both. We use a larger parallel size in the fill-up stage. Specifically, we set the allocation ratio $\rho$ between sequence fill-up and refinement to 0.5. Importantly, our method itself has no restriction on how steps are allocated; this constraint is introduced solely to ensure a fair comparison.

For our method, we adopt a cosine annealing scheduler for the refinement rate with $\gamma_{\max} = 0.8$ and $\gamma_{\min} = 0.4$, and increase the EoT penalty $\lambda_{eot}$ from 1.0 to 1.3 as the number of forward steps $T$ grows. For baselines, we use the recommended hyperparameters for ReMDM ($t_{on} = 0.55$, $t_{off} = 0.05$, $\alpha_{on} = 0.9$) and use the linear remasking scheduling function for RCR, which is reported to be optimal (He et al., 2025).

We follow the default prompts from the LM-Eval framework (Gao et al., 2024) and fix the generation length $L$ at 256. The total number of forward steps $T$ varies from 4 to 256 in powers of two, covering the scenarios from highly parallel to fully sequential decoding. All experiments are run on 8 NVIDIA H200 GPUs, and each data point is experimented with three random seeds for statistical significance.

## 4.2 MAIN RESULTS

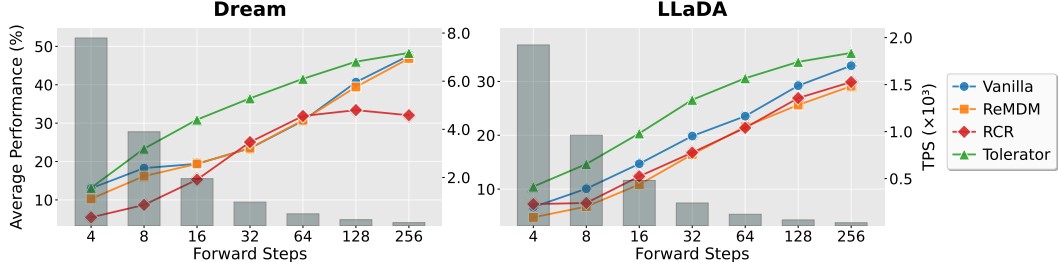

Figure 2: **Performance-Efficiency Trade-Off for Different Decoding Methods.** This figure illustrates the performance of different methods under varying parallel sizes. Gray bars represent generation throughput (tokens per second, TPS). Colored lines show average performance across five benchmarks as forward step $T$ varies.

To systematically evaluate the effectiveness of our approach across varying degrees of parallelism and task diversity, we conduct experiments on five standard benchmarks with forward steps $T$ ranging from 4 to 256. As illustrated in Figure 2, our method consistently improves performance under different parallel decoding configurations. On both Dream and LLaDA, we observe substantial gains in a large parallelism range (forward steps $T$ from 4 to 256), with average percentage score increasing from 29.0 to 34.6 (relatively +17.9%) and from 21.3 to 24.5 (+15.3%) compared to the strongest baseline in our experiments. These results indicate that our approach does not overfit to a specific parallel setting, but instead induces a consistent improvement in the quality–efficiency trade-off curve against all baselines. Moreover, as shown in Figure 3, performance gains generalize

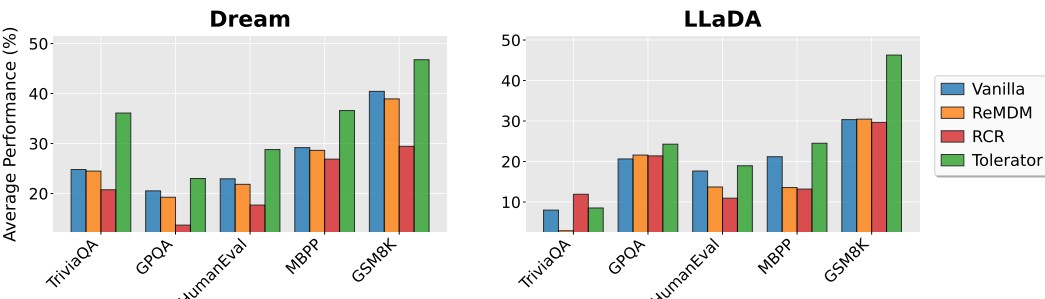

Figure 3: **Performance across different benchmarks for different decoding methods.** This figure presents the performance of various methods under different benchmarks. Colored bars represent average performance across different forward steps ($T$).

across tasks compared to the tested baselines: for example, on Dream, the average percentage score increases from 24.8 to 36.1 (+45.16%) on TriviaQA. While on LLaDA, it rises from 30.46 to 46.28 (+51.91%) on GSM8K. Collectively, these findings highlight the robustness and broad applicability of our method as a general enhancement for diffusion large language models. Detailed results for each task and forward steps can be found in Table 1.

### 4.3 ABLATION STUDIES

To analyze the effect of different components in our decoding strategy, we conduct ablation studies using GPQA (Rein et al., 2024) and GSM8K (Cobbe et al., 2021). In particular, we study the effectiveness of (1) token-level cross-validation refinement, (2) EoT Penalty, and (3) the annealing of refinement rate.

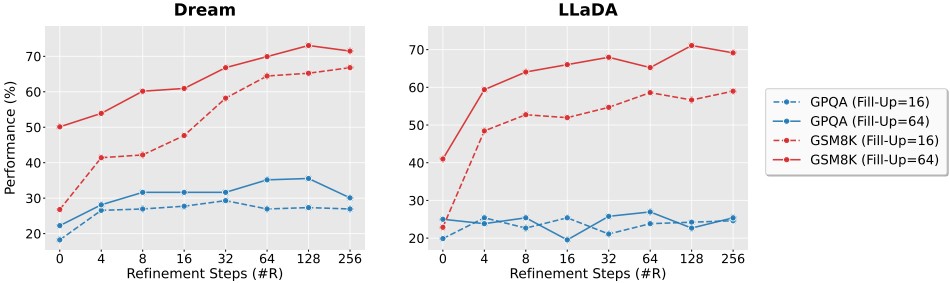

Figure 4: **Ablation Studies on Different Steps of Refinement Stage.** Red lines represent the results from GSM8K and blue lines represent GPQA. The solid lines stand for the results of 64 fill-up steps and the dashed lines for the results of 16 fill-up steps.

**Cross-Validation Refinement.** To isolate the role of the refinement stage, we focus on settings where the fill-up part is fixed, and then vary the degree of cross-validation refinement applied. Concretely, we fix the number of generation steps to either 16 or 64, and then allocate different amounts of refinement ranging from very few steps to nearly converged refinement. Specifically, we experiment with applying 4, 8, 16, 32, 64, 128, and 256 refinement steps.

As shown in Figure 4, in most cases, the performance curve with respect to refinement steps exhibits an increasing trend. This indicates that increasing the number of refinement steps—especially the initial steps—consistently improves the generation quality of dLLM. Therefore, the introduction of refinement can significantly enhance model performance, even with only a few steps.

**EoT penalty.** To isolate the impact of the EoT penalty, we fix the fill-up and refinement configurations and vary only the penalty coefficient $\lambda_{\text{eot}}$. Specifically, we vary $\lambda_{\text{eot}}$ from 1.0 to 1.3 while keeping the number of forward step $T$ fixed at 32 and 128. We find that applying non-trivial $\lambda_{\text{eot}}$ consistently improves generation quality, with notable gains at $\lambda_{\text{eot}} = 1.1$, 1.2, and 1.3 (+23.2%, +28.4%, +23.9% relatively). This is because the EoT penalty typically encourages longer fill-up sequence: although these drafts may not always be fully correct, they tend to contain more information

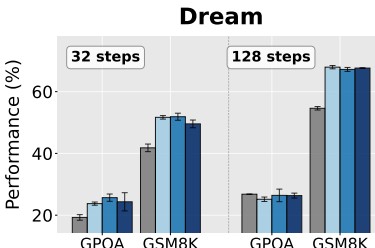
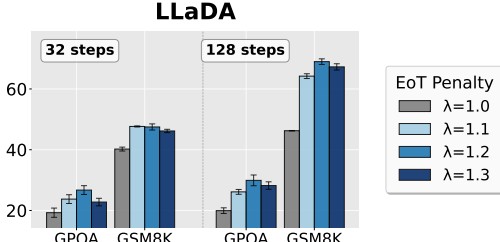

Figure 5: **Ablation Studies of EoT Penalty.** We fix the fill-up and refinement configurations while varying $\lambda_{\text{eot}}$ from 1.0 to 1.3, with results shown for 32 and 128 forward step $T$. Across most tasks, introducing an appropriate EoT penalty substantially improves generation quality.

overall. During refinement, the useful content can be preserved and amplified while the incorrect parts are likely to be corrected. Overall, these results demonstrate that explicitly regularizing the end-of-sequence token is a simple yet highly effective enhancement for our method.

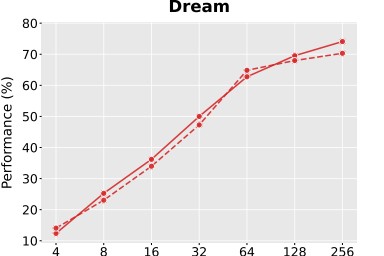
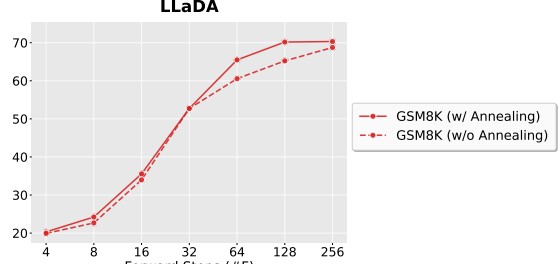

Figure 6: **Ablation Studies on Annealing Scheduling of Refinement Rate.** The solid lines represent the performance-forward step curve with the annealing strategy and the dashed lines represent the curves without the mechanism.

**Refinement Rate Annealing.** To assess the benefit of annealing schedule of refinement rate, we compare refinement with and without the cosine scheduler. We vary the number of forward steps from 4 up to 256 while keeping other parameters fixed, and report the resulting performance for both configurations.

The purpose of annealing is that, as the refinement process progresses, the overall quality of the generated sequence gradually improves. Consequently, fewer modifications are required to maintain stability and consistency. As illustrated in the Figure 6, the solid line is above the dashed line in most cases, demonstrating that the model with annealing outperforms its counterpart without it.

Overall, these ablations demonstrate that all three design choices contribute to the final performance. Exact numerical results can be found in Appendix B.

## 5 DISCUSSION

### 5.1 WHY OUR STRATEGY IS GOOD FOR LARGE PARALLEL SIZES?

We observe that our method achieves greater improvements when the parallel size is larger, i.e., when the forward step is smaller than the sequence length and multiple tokens are decoded simultaneously.

One key reason may lie in the visibility constraint during parallel decoding: tokens generated within the same step cannot attend to each other, which often leads to local inconsistencies. This phenomenon is even more noticeable with larger parallel sizes. Our token-level cross-validation process helps to mitigate this issue. During cross-validation, tokens filled up in the same step can be validated such that one serves as context (or validator) while another serves as the validation target. This mechanism enables tokens that were originally invisible to each other to interact directly—for

example, when validating token A, token B (from the same step) can now be used as part of the context. Such interactions promote coherence among simultaneously decoded tokens. By repeating this process across multiple rounds, inconsistencies introduced by parallel decoding are progressively reduced, resulting in more coherent sequences overall.

In contrast, when the forward step equals the sequence length (i.e., non-parallel decoding with one token per step), every token naturally conditions on all previously accepted tokens. Since there is no within-step invisibility, the inconsistency problem does not arise, and thus the potential benefit of our method is relatively limited in this scenario.

## 5.2 LIMITATIONS

**Format Stability.**   While our method achieves consistent improvements across a range of benchmarks, the gains are relatively smaller on code generation tasks such as HumanEval and MBPP. These tasks are highly format-sensitive, where even minor deviations in syntax or structure can make an otherwise correct solution invalid. Since our refinement process operates at the token level without explicit structural constraints, it can occasionally disrupt the formatting of well-formed code. This suggests a potential limitation when applying our strategy to domains where strict output format is essential. This limitation is also observed in methods like RCR (He et al., 2025), which need to do more remasking than vanilla generation, thereby disrupting the formatting of the sequence.

**Lack of Natural Convergence.**   In iterative sequence–refinement methods, a common stopping rule is natural convergence, which means the sequence remains unchanged after an iteration. However, with current approaches, even when we allow a large number of refinement steps, the model keeps making edits, even often unrelated to the final answer. As a result, the process often fails to naturally converge.

## 6 CONCLUSION

In this work, we revisited a key limitation of diffusion large language models (dLLMs): once a token is accepted during decoding, it is typically fixed and cannot be revised, causing early mistakes to persist and propagate through subsequent iterations. To address this, we proposed TOLERATOR, a training-free decoding strategy that explicitly decouples decoding into fill-up and refinement stages. By first generating a coarse draft and then iteratively remasking and decoding tokens with the token-level cross-validation principle, TOLERATOR enables more systematic and effective error correction than prior approaches.

Through extensive experiments on five benchmarks spanning natural language understanding, code generation, and mathematical reasoning, we showed that TOLERATOR consistently improves over baselines under the same forward step budgets. Beyond empirical gains, our results highlight that decoding strategy is not merely an implementation choice, but a crucial component that influences the overall performance of dLLMs.

## ETHICS STATEMENT

All datasets used in this work (TriviaQA (Joshi et al., 2017), GPQA (Rein et al., 2024), GSM8K (Cobbe et al., 2021), HumanEval (Chen et al., 2021), MBPP (Austin et al., 2021b)) are publicly available academic benchmarks that do not contain personally identifiable or sensitive information. Our study focuses on improving inference in discrete diffusion language models and does not involve the collection of new human subject data. We acknowledge that large language models may generate incorrect or misleading content, and that code generation models can potentially produce insecure or faulty programs. Our method does not eliminate these risks, and users should exercise caution when deploying such systems in high-stakes scenarios. The potential societal benefits of our work include improved decoding performance of diffusion large language models. This research was conducted in accordance with the ICLR Code of Ethics. The authors take full responsibility for all analyses and conclusions presented in this paper.

## REPRODUCIBILITY STATEMENT

We have taken several steps to ensure the reproducibility of our results. Our experiments were conducted on two representative open-source discrete diffusion language models: Dream-v0-Instruct-7B (Ye et al., 2025b) and LLaDA-8B-Instruct (Nie et al., 2025). We evaluate across five widely used public benchmarks—TriviaQA (Joshi et al., 2017), GPQA (Rein et al., 2024), GSM8K (Cobbe et al., 2021), HumanEval (Chen et al., 2021), and MBPP (Austin et al., 2021b). For all methods, we adopt the same model backbones, zero-shot setting, and equalized computational budgets to guarantee fairness. Reported results are averaged over 3 random seeds, and exact numerical results for both main experiments and ablations are provided in the appendix. We detail hyperparameter configurations in Section 4.1, including scheduler settings, penalty coefficients, and baseline parameters (ReMDM (Wang et al., 2025) and RCR (He et al., 2025)). Code, configuration files, and data preprocessing scripts are made anonymously available to facilitate replication. With the provided code and instructions, our results can be reproduced using 8×H200 GPUs or equivalent hardware.

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

## A USE OF LLMs DISCLOSURE

We disclose the following uses of large language models in the preparation of this work. GPT-5 (OpenAI, 2025) was employed solely to assist with language polishing and improving the readability of the manuscript. In addition, Claude Code (Anthropic, 2025) was used as a coding assistant to generate and debug experimental scripts. At no point did LLMs contribute to the core research ideas, methodology, or interpretation of results. All scientific contributions, analyses, and conclusions remain the responsibility of the authors. Outputs produced by LLMs were carefully reviewed and revised where necessary to ensure accuracy and integrity.

## B EXPERIMENTAL DETAILS

### B.1 MAIN EXPERIMENT

In the main text, we present line and bar plots to highlight overall trends and comparisons on different tasks and forward step $T$. For completeness, Appendix B reports the exact numerical results of our main experiments in tabular form, which allow for more precise inspection and direct comparison across different methods and settings.

Table 1: **Main Experiment Results**. Performance of Dream and LLaDA across five standard benchmarks under different numbers of forward steps. Highest values for specific task and model are **bold**.

| Model | Method | TriviaQA | | | | | | |
|---|---|---|---|---|---|---|---|---|
| | | #F=4 | #F=8 | #F=16 | #F=32 | #F=64 | #F=128 | #F=256 |
| Dream | Vanilla | 23.08±0.01 | 23.22±0.02 | 23.16±0.03 | 23.24±0.02 | 23.51±0.01 | 28.08±0.03 | 29.32±0.03 |
| | ReMDM | 22.11±1.17 | 22.94±0.28 | 22.87±0.10 | 22.94±0.16 | 23.27±0.32 | 27.98±0.41 | 29.26±0.37 |
| | RCR | 15.63±0.23 | 14.53±0.12 | 15.02±0.13 | 17.68±0.13 | 18.92±0.27 | 26.81±0.34 | 36.64±0.42 |
| | Tolerator | **27.78**±0.29 | **31.61**±0.11 | **33.76**±0.19 | **35.98**±0.16 | **40.61**±0.16 | **42.46**±0.22 | **40.47**±0.16 |
| LLaDA | Vanilla | 0.19±0.02 | 0.65±0.03 | 2.13±0.01 | 4.63±0.02 | 9.36±0.06 | 16.25±0.02 | 22.76±0.01 |
| | ReMDM | 0.25±0.02 | 0.43±0.01 | 1.08±0.02 | 1.82±0.03 | 3.05±0.06 | 5.43±0.03 | 8.24±0.02 |
| | RCR | 0.09±0.01 | 0.80±0.01 | **4.44**±0.01 | **8.62**±0.01 | **16.08**±0.02 | **24.04**±0.01 | **29.30**±0.01 |
| | Tolerator | **0.99**±0.01 | **1.86**±0.08 | 3.52±0.09 | 6.19±0.06 | 10.94±0.10 | 16.46±0.09 | 19.72±0.14 |

| Model | Method | GPQA | | | | | | |
|---|---|---|---|---|---|---|---|---|
| | | #F=4 | #F=8 | #F=16 | #F=32 | #F=64 | #F=128 | #F=256 |
| Dream | Vanilla | **10.27**±0.59 | 17.04±0.34 | 18.23±0.56 | 20.91±1.10 | 22.25±0.93 | 27.01±0.80 | 27.98±0.85 |
| | ReMDM | 7.44±1.05 | 15.92±1.03 | 17.93±0.13 | 19.27±0.90 | 22.62±1.12 | 23.36±0.72 | 28.20±0.68 |
| | RCR | 1.79±0.21 | 3.12±0.13 | 7.81±0.11 | 14.06±0.28 | 25.00±0.32 | 24.78±0.41 | 19.20±0.43 |
| | Tolerator | 8.11±1.05 | **17.19**±0.97 | **22.84**±0.13 | **26.71**±1.45 | **26.93**±1.10 | **29.91**±1.77 | **29.32**±1.23 |
| LLaDA | Vanilla | 10.79±1.58 | 13.47±1.58 | 19.87±1.39 | 23.88±0.67 | 25.00±1.02 | 25.37±0.46 | 26.04±0.13 |
| | ReMDM | 9.60±0.89 | 16.67±0.13 | **23.66**±1.18 | 24.70±1.01 | 25.74±0.68 | 25.82±1.01 | 24.93±0.13 |
| | RCR | 20.46±0.46 | 18.45±0.13 | 19.05±0.13 | 18.97±0.22 | 21.80±0.13 | 26.19±0.13 | 24.78±0.13 |
| | Tolerator | **20.76**±1.46 | **20.76**±1.46 | 22.47±1.45 | **25.67**±1.18 | **27.01**±1.56 | 26.41±2.03 | **26.86**±1.49 |

| Model | Method | HumanEval | | | | | | |
|---|---|---|---|---|---|---|---|---|
| | | #F=4 | #F=8 | #F=16 | #F=32 | #F=64 | #F=128 | #F=256 |
| Dream | Vanilla | **8.13**±0.35 | 13.41±0.00 | 11.79±0.35 | 12.80±0.61 | 26.02±0.35 | 37.80±0.61 | **50.61**±0.00 |
| | ReMDM | 2.03±0.70 | 9.35±0.35 | 12.20±0.00 | 13.21±0.35 | 30.49±0.37 | 38.82±0.70 | 50.20±0.93 |
| | RCR | 1.22±0.24 | 8.54±0.31 | 8.54±0.31 | 22.56±0.45 | 30.49±0.37 | 26.22±0.28 | 26.22±0.28 |
| | Tolerator | 4.88±1.06 | **17.89**±1.27 | **27.03**±2.54 | **30.89**±1.37 | **33.03**±2.21 | **40.24**±0.81 | 47.56±0.61 |
| LLaDA | Vanilla | 9.55±0.35 | **14.23**±0.35 | 15.24±1.22 | 15.45±1.27 | 18.29±1.40 | 23.68±2.54 | **27.13**±0.30 |
| | ReMDM | 4.88±1.22 | 6.10±0.00 | 8.13±1.27 | 10.37±1.22 | 18.09±3.07 | 22.76±0.70 | 25.61±3.23 |
| | RCR | **9.96**±0.35 | 5.08±0.93 | 7.52±0.35 | 7.93±0.35 | 11.99±0.35 | 15.85±0.84 | 18.29±0.31 |
| | Tolerator | 7.52±1.53 | 12.40±0.35 | **20.43**±1.40 | **23.58**±0.93 | **22.05**±0.77 | **24.19**±0.77 | 22.46±5.99 |

| Model | Method | MBPP | | | | | | |
|---|---|---|---|---|---|---|---|---|
| | | #F=4 | #F=8 | #F=16 | #F=32 | #F=64 | #F=128 | #F=256 |
| Dream | Vanilla | **14.40**±0.20 | 14.73±0.12 | 17.00±0.20 | 25.00±0.40 | 31.07±0.12 | 45.13±0.31 | **56.93**±0.83 |
| | ReMDM | 8.80±0.53 | 14.67±0.31 | 15.93±0.12 | 26.00±0.20 | 33.13±0.70 | 45.27±0.64 | 56.60±0.35 |
| | RCR | 4.80±0.84 | 10.40±0.71 | 23.60±0.55 | 29.00±0.29 | 36.00±0.36 | 42.60±0.43 | 41.73±0.12 |
| | Tolerator | 10.53±1.01 | **25.13**±0.64 | **35.00**±0.80 | **41.07**±2.91 | **44.40**±1.20 | **48.47**±1.55 | 51.53±0.76 |
| LLaDA | Vanilla | **9.53**±0.81 | 14.40±0.40 | 13.40±0.69 | 17.73±0.12 | 24.07±0.64 | 31.27±0.81 | 37.87±0.64 |
| | ReMDM | 1.53±0.31 | 2.33±0.58 | 4.53±0.42 | 10.53±0.12 | 17.27±0.64 | 23.33±1.72 | 35.47±0.90 |
| | RCR | 0.60±0.40 | 3.47±0.46 | 10.00±0.20 | 13.33±0.12 | 15.93±0.46 | 22.27±0.31 | 26.73±0.12 |
| | Tolerator | 5.53±1.03 | **16.00**±0.87 | **22.73**±1.03 | **25.60**±0.69 | **29.27**±0.42 | **33.87**±0.81 | **38.53**±1.50 |

| Model | Method | GSM8K | | | | | | |
|---|---|---|---|---|---|---|---|---|
| | | #F=4 | #F=8 | #F=16 | #F=32 | #F=64 | #F=128 | #F=256 |
| Dream | Vanilla | 9.22±0.46 | 23.07±0.04 | 26.79±0.12 | 35.36±0.09 | 50.11±0.00 | 65.38±0.12 | **73.10**±0.06 |
| | ReMDM | 11.02±0.64 | 18.12±0.59 | 27.98±0.00 | 35.91±0.09 | 47.81±0.09 | 61.66±0.16 | 70.00±0.44 |
| | RCR | 3.79±0.25 | 6.90±0.27 | 21.46±0.33 | 42.00±0.41 | 48.90±0.38 | 46.55±0.29 | 36.47±0.35 |
| | Tolerator | **14.40**±0.59 | **24.92**±1.22 | **35.96**±1.45 | **47.66**±0.18 | **62.80**±0.90 | **68.99**±0.92 | 72.61±0.46 |
| LLaDA | Vanilla | 3.23±0.18 | 7.58±0.35 | 22.87±0.70 | 37.55±0.42 | 40.99±0.61 | 49.46±0.74 | 50.75±0.57 |
| | ReMDM | 7.46±0.16 | 8.24±1.10 | 16.83±0.40 | 34.77±0.61 | 43.85±0.24 | 50.77±0.24 | 51.33±0.81 |
| | RCR | 4.93±0.32 | 9.29±0.70 | 20.81±0.27 | 35.03±0.54 | 41.09±0.64 | 46.17±0.32 | 50.34±0.96 |
| | Tolerator | **17.49**±0.43 | **22.24**±0.90 | **32.58**±1.20 | **51.88**±1.14 | **63.66**±0.23 | **67.20**±0.64 | **68.89**±1.05 |

## B.2 Ablation Studies

Similarly, we present the exact numerical results our further analysis on different components in tabular form above.

Table 2: **Performance under different refinement steps (#R) with fixed fill-up stage steps (16 or 64).** Results are reported for both Dream-Instruct and LLaDA on GPQA and GSM8K.

| Fill-Up Steps | Model | Task | #R=0 | #R=4 | #R=8 | #R=16 | #R=32 | #R=64 | #R=128 | #R=256 |
|---|---|---|---|---|---|---|---|---|---|---|
| **16** | Dream | GPQA | 18.23 | 26.56 | 26.95 | 27.73 | 29.30 | 26.95 | 27.34 | 26.95 |
| | Dream | GSM8K | 26.79 | 41.41 | 42.19 | 47.66 | 58.20 | 64.45 | 65.23 | 66.80 |
| | LLaDA | GPQA | 19.87 | 25.39 | 22.66 | 25.39 | 21.09 | 23.83 | 24.22 | 24.61 |
| | LLaDA | GSM8K | 22.87 | 48.44 | 52.73 | 51.95 | 54.69 | 58.59 | 56.64 | 58.98 |
| **64** | Dream | GPQA | 22.25 | 28.12 | 31.64 | 31.64 | 31.64 | 35.16 | 35.55 | 30.08 |
| | Dream | GSM8K | 50.11 | 53.91 | 60.16 | 60.94 | 66.80 | 69.92 | 73.05 | 71.48 |
| | LLaDA | GPQA | 25.00 | 23.83 | 25.39 | 19.53 | 25.78 | 26.95 | 22.66 | 25.39 |
| | LLaDA | GSM8K | 40.99 | 59.38 | 64.06 | 66.02 | 67.97 | 65.23 | 71.09 | 69.14 |

Table 3: **Performance with different values of the EoT penalty coefficient $\lambda_{\text{eot}}$ (1.0–1.3) under fixed fill-up and refinement configurations.** Evaluated on GPQA and GSM8K with Dream-Instruct and LLaDA. Reported as mean (± variance) over 3 seeds.

| Forward Steps | Model | Task | $\lambda_{\text{eot}} = 1.0$ | 1.1 | 1.2 | 1.3 |
|---|---|---|---|---|---|---|
| **32** | Dream | GPQA | 19.27 ±0.90 | 23.74 ±0.52 | 25.67 ±1.18 | 24.33 ±2.95 |
| | Dream | GSM8K | 41.80 ±1.23 | 51.68 ±0.57 | 51.88 ±1.14 | 49.56 ±1.25 |
| | LLaDA | GPQA | 19.27 ±1.52 | 23.74 ±1.44 | 26.71 ±1.45 | 22.77 ±1.24 |
| | LLaDA | GSM8K | 40.21 ±0.64 | 47.66 ±0.18 | 47.49 ±1.01 | 46.17 ±0.55 |
| **128** | Dream | GPQA | 26.79 ±0.00 | 25.15 ±0.72 | 26.41 ±2.03 | 26.34 ±0.80 |
| | Dream | GSM8K | 54.61 ±0.56 | 67.93 ±0.53 | 67.20 ±0.64 | 67.63 ±0.13 |
| | LLaDA | GPQA | 19.94 ±0.93 | 26.12 ±0.80 | 29.91 ±1.77 | 28.20 ±1.23 |
| | LLaDA | GSM8K | 46.25 ±0.08 | 64.22 ±0.79 | 68.99 ±0.92 | 67.27 ±1.03 |

Table 4: **Performance of LLaDA and Dream models on GSM8K across different forward numbers (#F).** We compare refinement with and without annealing.

| Model | Task | Setting | #F=4 | #F=8 | #F=16 | #F=32 | #F=64 | #F=128 | #F=256 |
|---|---|---|---|---|---|---|---|---|---|
| LLaDA | GSM8K | With Annealing | 20.31 | 24.22 | 35.55 | 52.73 | 65.49 | 70.18 | 70.31 |
| | | Without Annealing | 19.92 | 22.66 | 33.98 | 52.73 | 60.55 | 65.23 | 68.75 |
| Dream | GSM8K | With Annealing | 12.37 | 25.26 | 36.20 | 50.00 | 62.76 | 69.53 | 74.09 |
| | | Without Annealing | 14.06 | 23.05 | 33.98 | 47.27 | 64.84 | 67.97 | 70.31 |

## C  QUALITATIVE EXAMPLES

In addition to quantitative results, we provide qualitative examples to illustrate how token-level cross-validation can effectively correct errors in accepted tokens. In this example, we set fill-up step and refinement step both to 16.

As shown in Figure 7 through Figure 10, the initial filled sequences often contain both grammatical inconsistencies (e.g., redundant phrases such as "the number the number") and semantic errors (e.g., producing an incorrect result such as 88,000000). Through iterative refinement, inconsistent tokens are either modified or removed, while more appropriate tokens are introduced. This process progressively reduces grammatical and semantic errors, ultimately yielding the correct answer (e.g., 8000 copies).

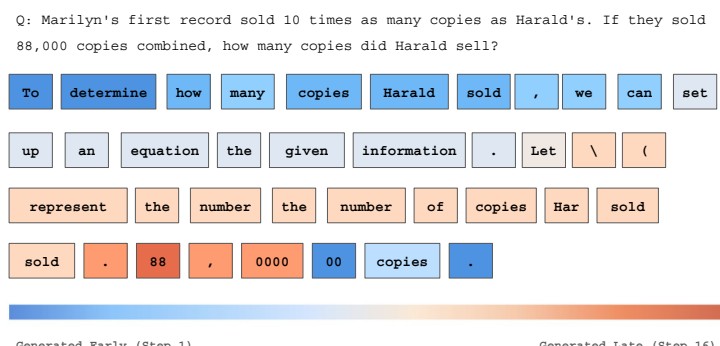

Figure 7: **Output of Fill-Up Stage**.We use colors fading from blue to red to demonstrate the order of decoding.Using fill-up and refinement steps =16, the special special tokens like [EoT] are not shown.

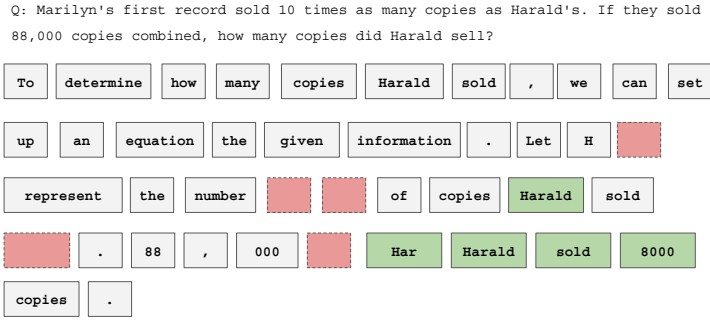

Figure 8: **Sequence after 1 Iteration of Refinement.** Red dashed boxes represent deleted tokens while green boxes represent added tokens in current iteration.

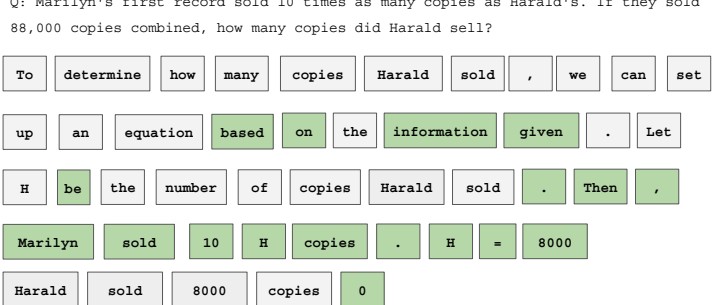

Figure 9: **Sequence after 8 Iteration of Refinement**. Red dashed boxes represent deleted tokens while green boxes represent added tokens in the current iteration.

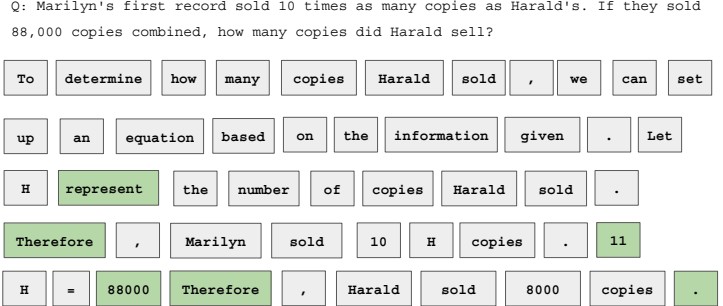

Figure 10: **Sequence after 16 Iteration of Refinement**. Red dashed boxes represent deleted tokens while green boxes represent added tokens in the current iteration.

