# OpenReview forum: "Finish First, Perfect Later: Test-Time Token-Level Cross-Validation for Diffusion Large Language Models"
_ICLR.cc/2026/Conference — ICLR 2026 Conference Withdrawn Submission_

### Official Review · Reviewer_Chej · 2025-10-31

**Soundness:** 2
**Presentation:** 2
**Contribution:** 2
**Rating:** 4
**Confidence:** 4

**Summary:**

This paper tackles a key limitation in dLLM decoding, where committed tokens cannot be revised, causing early errors to persist. To address this, the authors propose TOLERATOR (Token-Level Cross-Validation Refinement), a training-free strategy that first generates a complete draft using standard dLLM decoding (Sequence Fill-Up) and then iteratively refines it (Cross-Validation Refinement) by remasking subsets of tokens and re-predicting them using the remaining tokens as context.

**Strengths:**

1. The paper accurately identifies the core flaw in current dLLM decoding strategies: the irreversibility of errors. As dLLMs gain attention as alternatives to autoregressive (AR) models, addressing the trade-off between decoding efficiency and quality is highly valuable.
2. Code is provided to facilitate reproducibility.

**Weaknesses:**

1. The main issue with the paper lies in its practical performance: recent methods that reduce sampling steps (e.g., arXiv:2505.22618, arXiv:2506.10848, arXiv:2507.18578) can accelerate inference while maintaining quality, yet the proposed approach often underperforms vanilla decoding in many scenarios, even with the same computational budget (Table 1).
2. Some baselines appear unusually low. For example, LLaDA on GSM8K achieves only around 50%, which seems implausibly low.
3. Experiments show that the EoT penalty provides substantial improvements, but it is not a core component of the proposed method, which further diminishes the practical impact of the main approach.

**Questions:**

See Weaknesses.

---

### Official Review · Reviewer_d6B7 · 2025-10-31

**Soundness:** 3
**Presentation:** 3
**Contribution:** 2
**Rating:** 4
**Confidence:** 4

**Summary:**

This paper introduces TOLERATOR (Token-Level Cross-Validation Refinement), a training-free decoding strategy for diffusion large language models (dLLMs). Compared to conventional dLLM, decoding fixes tokens once they’re accepted, which causes early mistakes to persist,  TOLERATOR uses a two-stage decoding process:

	1.	Fill-up stage: generate a coarse draft with high parallelism (vanilla diffusion decoding).

	2.	Refinement stage: iteratively remask and regenerate subsets of tokens, where some tokens act as “context” and others as “validation targets,” allowing mutual correction through cross-validation.

Experiments on five benchmarks—TriviaQA, GPQA, GSM8K, MBPP, and HumanEval—show consistent performance improvements (e.g., +17.9% for Dream and +15.3% for LLaDA) under the same computational budget.

**Strengths:**

1. The motivation of the paper is clear and it proposes a conceptually elegant and general solution.

2. The authors have done comprehensive evaluation across five benchmarks and two leading dLLMs (Dream and LLaDA).  Consistent improvements across tasks and varying degrees of parallel decoding (T=4–256) are demonstrating robustness of the proposed method.

3. The authors evaluate contributions of refinement steps, end-of-text (EoT) penalty, and annealing schedule, which shows that each design choice contributes to performance gains.

4. The paper also provides a conceptual explanation for why cross-validation helps especially with large parallel sizes, to help the readers understand what's the insights when tokens in the same step can’t see each other.

5. The writing and structure of the paper is overall clear, plus anonymous code link provided, with reproducibility details.

**Weaknesses:**

1. The paper lacks a formal justification for why cross-validation improves the likelihood or convergence properties of the diffusion process. Moreover, there is no analytical comparison with prior probabilistic remasking approaches (e.g., ReMDM (Wang et al., 2025)).

2. Evaluation mainly focuses on standard text and code tasks; lacks analysis on more complex scenarios such long-context generation (e.g., BigCodeBench), where diffusion LMs may differ most from AR models.

3. While authors claim equal “forward-step” cost, the two-stage process may incur subtle computational overhead (e.g., due to multiple refinement passes). Clarity on real wall-clock time or GPU-hour parity would strengthen fairness.

4. As acknowledged by the authors, refinement can fail to converge naturally, which raises questions about how to determine optimal refinement steps or stopping criteria.

5. Missing prior works discussions. WINO (arXiv:2507.18578) already demonstrated training-free iterative token verification and revocation for dLLMs, enabling backward revision of previously accepted tokens. TOLERATOR’s idea of “remasking accepted tokens during decoding” is conceptually aligned, differing only in timing (post-hoc refinement vs. in-process correction).  The authors need to clarify the differences in the paper and compare if necessary.

**Questions:**

1. Are there heuristics to detect convergence (e.g., minimal token change ratio), or could reinforcement learning be applied to decide refinement length?

2. Can cross-validation refinement be expressed as an approximate consistency regularization or ensemble expectation over token subsets?

3. Could TOLERATOR’s annealed refinement be integrated on top of related methods, such as dLLM cache, WINO etc.?

---

### Official Review · Reviewer_gQk7 · 2025-11-02

**Soundness:** 2
**Presentation:** 3
**Contribution:** 2
**Rating:** 4
**Confidence:** 5

**Summary:**

This paper targets a core limitation of discrete diffusion LLM decoding, irreversible early token acceptances that propagate errors, and proposes TOLERATOR, a training-free, model-agnostic decoding strategy that explicitly decouples generation into (i) sequence fill-up and (ii) cross-validation refinement. In Stage I, the model performs standard parallel fill-up but regularizes premature termination via an End-of-Text penalty; in Stage II, it iteratively remasks a random subset $S^{(k)}$ of positions at rate $\gamma_k$ and redecodes them conditioned on the fixed remainder, so tokens alternately serve as validator and validation target (“token-level cross-validation”). The refinement rate follows a cosine schedule, while total forward steps $T$ are split by ratio $\rho$ between fill-up and refinement under a matched compute budget to baselines. Across Dream-v0-Instruct-7B and LLaDA-8B-Instruct on TriviaQA, GPQA, HumanEval, MBPP, and GSM8K, TOLERATOR delivers consistent gains over vanilla, ReMDM, and RCR—on average +17.9% (Dream) and +15.3% (LLaDA)—robust to parallelism levels, with strong improvements at larger parallel sizes where within-step “invisibility” creates local inconsistencies. Ablations isolate each component’s effect (more refinement steps help; $\lambda_{\text{eot}}$ improves drafts; annealed $\gamma_k$ stabilizes convergence). Limitations are candidly discussed (smaller gains on format-sensitive code; lack of natural convergence). Contributions: a clear reframing of dLLM decoding as fill-up + token-level cross-validation; a simple EoT penalty and annealed remasking schedule; and broad empirical evidence that decoding algorithms, not only model training, materially improve dLLM quality-efficiency trade-offs.

**Strengths:**

- The paper reframes diffusion-LM decoding into a two-stage pipeline: fill-up followed by token-level cross-validation refinement. Instead of irreversibly accepting tokens step by step, the method remasks a subset at rate $\gamma_k$ so tokens alternately play validator and target roles—making error correction explicit. It's a simple but clever idea: training-free and model-agnostic cross-validation of token interactions, plus an End-of-Text penalty and cosine annealing schedule. This tackles a core limitation of prior decoding strategies with minimal algorithmic overhead.

- The experiments are promising; results span two models (Dream-7B, LLaDA-8B) and five benchmarks (TriviaQA, GPQA, HumanEval, MBPP, GSM8K) with three seeds. Gains are consistent and substantial (average +17.9% on Dream, +15.3% on LLaDA), especially at high parallelism where within-step "invisibility" hurts most. The ablations isolate each component: more refinement helps, $\lambda_{\text{eot}}$ improves drafts, and annealing stabilizes convergence. The analysis connects wins to the parallel architecture rather than hyperparameter tuning.

- The problem is crisply formalized, the method notation is precise (remasking sets $S^{(k)}$, annealed $\gamma_k$, EoT scaling), and compute accounting is transparent. Figures and tables cleanly separate throughput vs. accuracy; ablations tell a clear causal story; hyperparameters and their effects are documented for reproducibility. The narrative makes the intuition behind cross-validation at high parallelism obvious.

- The method is drop-in, training-free, and model-agnostic, so practitioners can immediately use it with existing diffusion LMs. The broad, budget-matched improvements and the token-level cross-validation framework elevate decoding from an implementation detail to a first-class algorithmic lever for diffusion LMs. This will likely influence both practitioners (who want quality without retraining) and researchers (exploring richer refinement schemes), and it strengthens the case for diffusion LMs as viable, fast alternatives to autoregressive decoding.

**Weaknesses:**

- EoT penalty
   §3.3 states the method "scale[s] down the logit of the EoT token by a factor $\lambda_{\text{eot}}>1$ before softmax," but the formula applies a probability rescale: $\tilde{p}(v)\propto \exp(z_v)/\lambda_{\text{eot}}$ for $v=[\text{EOT}]$. Dividing the post-exp term is equivalent to subtracting $\log\lambda_{\text{eot}}$ from the logit (an additive shift), not scaling the logit by $1/\lambda$.


- The paper equalizes "number of forward steps" ($T$), but per-step work differs across methods, so reporting FLOPs/token and throughput for each setting, and clarifying how per-step token counts compare across methods, may make more sense.


- “Consistent improvements” claim is not supported by Table 1
    There are multiple regimes where TOLERATOR underperforms a baseline:

    - Dream/GPQA, #F=4: Vanilla 10.27 vs. TOLERATOR 8.11.
    - Dream/HumanEval, #F=256: Vanilla 50.61 vs. TOLERATOR 47.56.
    - Dream/MBPP, #F=256: Vanilla 56.93 vs. TOLERATOR 51.53.
    - LLaDA/TriviaQA, #F=16–256: RCR dominates (e.g., #F=256: RCR 29.30 vs. TOLERATOR 19.72).
    - LLaDA/HumanEval, #F=256: Vanilla 27.13 vs. TOLERATOR 22.46.

    These contradict statements like “our method consistently improves under different parallel decoding configurations” (§4.2) and “consistent improvements over the baselines under the same computational budget” (Abstract).

-  §4.3 says refinement steps show an “increasing trend,” but Table 2 shows several non-monotonicities (e.g., Dream/GPQA 16 fill-up: 29.30 at #R=32 then 26.95 at #R=64; LLaDA/GPQA 64 fill-up: 26.95 at #R=64 then 22.66 at #R=128).


-  Annealing helps "in most cases," but also hurts noticeably.
    Table 4 shows Dream/GSM8K at #F=64: 62.76 (with) vs. 64.84 (without); LLaDA/GSM8K #F=128: 70.18 vs 65.23 (helps).

-  §4.2 mentions very large TriviaQA and GSM8K improvements; however, on LLaDA/TriviaQA the strongest baseline (RCR) dominates at higher #F.

-    §3.1/§3.3 allude to confidence/entropy but give no formula or thresholds.

- Notation wise, in §3.4 the paper presents:
   $$
   x_i^{(k)}=
     \begin{cases}
       [\text{MASK}], & i\in S^{(k)} \\
       x_i^{(k)}, & \text{otherwise}
     \end{cases}
   $$
   which defines $x^{(k)}$ in terms of itself. This should be a masked copy (e.g., $\tilde{x}^{(k)}$) formed from the previous iterate: $\tilde{x}^{(k)} = \text{Mask}(x^{(k)}; S^{(k)})$, followed by decoding to get $x^{(k+1)}$. Likewise, the next line conditions on the wrong object:
   $$
   y_i^{(k)}\sim p_\theta(\cdot\,|\,\tilde{x}^{(k)},\,\text{time})
   $$

**Questions:**

- In §3.3 the paper states that after the fill-up stage the sequence is fully unmasked:
   $$
   x^{(\rho T)}=\left(c_1,\dots,c_m,x^{(\rho T)}_{m+1},\dots,x^{(\rho T)}_L\right),\quad x_i^{(\rho T)}\neq[\text{MASK}]\ \forall i>m.
   $$
   But under the "vanilla" scheme in §3.1, each step accepts approximately $L/T$ tokens, so after only $\rho T < T$ steps there will still be masked tokens unless the acceptance schedule is altered. Either (i) $\rho=1$, (ii) the method accepts $>L/T$ tokens per step during fill-up (not described), or (iii) the claim that all masks are gone at step $\rho T$ is incorrect?


- With $\gamma_{\min}=0.4$, even the last refinement iteration remasks approximately $0.4(L-m)$ tokens, preventing fixed-point convergence by design (as also noted in §5.2). If "natural convergence" is a desideratum, why not anneal to $\gamma_{\min}\approx 0$ or a stopping rule $\mathbb{1}[x^{(k+1)}=x^{(k)}]$?

- §3.1/§3.3 allude to confidence/entropy, what are the thresholds?

---

### Official Review · Reviewer_SrS7 · 2025-11-04

**Soundness:** 1
**Presentation:** 2
**Contribution:** 1
**Rating:** 2
**Confidence:** 4

**Summary:**

This paper addresses a key limitation in current diffusion language models (DLLMs): once a token is generated, it is typically never revised, which can propagate early errors and limit overall quality. The authors propose a **two-stage test-time token-level cross-validation** approach to mitigate this issue.

In the first stage, a standard dLLM sampler generates a complete initial draft sequence starting from all masks. In the second stage, a subset of tokens is randomly remasked and regenerated using the remaining context. This remasking-refinement process is repeated multiple times under a **cosine annealing schedule**, where the remasking rate decreases over time to stabilize predictions.

Experiments demonstrate strong improvements, with relative gains of **17.9% on Dream** and **15.3% on LLaDA**, showing that the proposed method effectively enhances reasoning and consistency in diffusion language models without additional training.

**Strengths:**

1. The idea of remasking tokens is both timely and important for diffusion language models. While prior works have proposed alternative strategies to remask previously unmasked tokens, this paper provides compelling evidence that **repeated remasking of an initial draft**, generated via confidence-based unmasking samplers such as those used in LLaDA, can consistently improve performance.

2. The improvements are demonstrated across **five benchmarks** spanning natural language understanding, code generation, and mathematical reasoning, highlighting the generality of the proposed method.  The proposed reweighting of the *end-of-text (EoT)* token during draft generation is a simple yet effective idea that yields promising gains in generation quality.

**Weaknesses:**

1.  The abstract claims that once a token is accepted, it is no longer changed in diffusion language models. While this is true for earlier methods such as MDLM, MD4, and LLaDA, more recent works like **ReMDM** (https://openreview.net/pdf?id=IJryQAOy0p, NeurIPS 2025) and **ADLM** (https://openreview.net/pdf?id=E8adS5srds, NeurIPS 2025) already support token remasking. Although the introduction briefly discusses ReMDM, the abstract should clarify the challenges that remain in existing remasking algorithms and what specifically differentiates this work from these relevant baselines.

2. The distinction between the proposed approach and ReMDM should be discussed in detail. ReMDM includes a baseline where remasking can be disabled early and re-enabled later in the reverse process. With appropriate hyperparameter choices, the proposed method seems to reduce to a special case of ReMDM. The paper should explicitly describe how its algorithm decouples fill-up and refinement, and how it meaningfully differs from ReMDM’s stochastic remasking strategy.

3. In the iterative refinement stage, remasked tokens are predicted as $y_i^{(k)} \sim p_\theta(\cdot | x^{(k)}, k)$. Since the pretrained model conditions on discrete time steps $t$, the refinement index $k$ should not differ arbitrarily from these training-time values. It is unclear whether the proposed cosine-annealed refinement rate can deviate from the pretrained denoising schedule. The authors should clarify this apparent mismatch.

4. The reported base performance of LLaDA (e.g., 30.46 vs. 70.3 in the original paper) differs significantly from the published results. The proposed method achieves 46.28, but it remains unclear whether improvements persist under identical settings (e.g. with **in-context examples**, which LLaDA shows can strongly improve accuracy). The evaluation setup should be made consistent for a fair comparison.

5. While increasing refinement steps generally improves results, GPQA results on LLaDA show little to no gain. The authors should analyze why performance plateaus in this case and whether this limitation arises from dataset characteristics, the base model, or the proposed remasking strategy itself.

6. The ablation on refinement-rate annealing shows minimal gains. In some cases (e.g., Dream dataset at forward steps 4, 64 and in the neighborhood), the model without refinement even outperforms the refined version. This raises doubts about the claimed importance of design choices, suggesting that their contribution might be overstated.

7. The claim that tokens unmasked at the same time step do not interact contradicts prior work. ADLM demonstrates that a two-stage process (first generating soft tokens, then remasking/unmasking) allows token interactions through soft samples, leading to improved remasking. ReMDM combined with ADLM outperforms ReMDM with MDLM (https://s-sahoo.com/mdlm/). The paper should discuss this distinction.

8. Lines 452-456:  The authors claim a limitation based on remasking at the end of the process but do not provide supporting empirical evidence. Concurrent works at ICLR 2026 show that final-stage remasking has negligible effect on LLaDA base models.
9. The conclusion suggests that this paper is the first to introduce remasking in diffusion language models. However, earlier works (ReMDM, ADLM) already explore token-level remasking. The conclusion should be revised to accurately position this paper’s contribution relative to prior remasking works.

**Questions:**

Please see the weakness section above.

---

### Author Response · Authors · 2025-11-13

Dear Reviewers,

We sincerely appreciate the time and effort you devoted to reviewing our submission. Your comments have helped us identify several important weaknesses in our work. In particular, we realized that our explanation of some unintuitive experimental results was not sufficiently clear, and that our distinction between cultural factors and the baselines was not made carefully enough.

Given the extent of revisions and additional analyses required to properly address these issues, we have decided to withdraw this submission from the current ICLR review cycle. We are very grateful for your constructive feedback, which will significantly guide us in improving the paper for a future submission.

Thank you again for your thoughtful and detailed reviews.

Best regards,
Authors

---

### Note · Authors · 2025-11-13

**Comment:**

Dear Reviewers,

We sincerely appreciate the time and effort you devoted to reviewing our submission. Your comments have helped us identify several important weaknesses in our work. In particular, we realized that our explanation of some unintuitive experimental results was not sufficiently clear, and that our distinction between cultural factors and the baselines was not made carefully enough.

Given the extent of revisions and additional analyses required to properly address these issues, we have decided to withdraw this submission from the current ICLR review cycle. We are very grateful for your constructive feedback, which will significantly guide us in improving the paper for a future submission.

Thank you again for your thoughtful and detailed reviews.

Best regards, Authors

**Withdrawal Confirmation:**

I have read and agree with the venue's withdrawal policy on behalf of myself and my co-authors.